# Racial/Ethnic Differences in the Pharmacokinetics of Antipsychotics: Focusing on East Asians

**DOI:** 10.3390/jpm12091362

**Published:** 2022-08-24

**Authors:** Shih-Ku Lin

**Affiliations:** 1Department of Psychiatry, Linkou Chang Gung Memorial Hospital, Taoyuan 333, Taiwan; sklin@tpech.gov.tw; Tel.: +886-2-27263141 (ext. 1263); 2Department of Psychiatry, Psychiatry Center, Taipei City Hospital, Taipei 110, Taiwan

**Keywords:** pharmacokinetic, cytochrome p450, antipsychotics, racial difference, therapeutic drug monitoring

## Abstract

Empirical clinical studies have suggested that East Asian patients may require lower dosages of psychotropic drugs, such as antipsychotics, lithium, and antidepressants, than non-Asians. Both the pharmacokinetic and pharmacodynamic properties of a drug can affect the clinical response of an illness. The levels of antipsychotics used for the treatment of schizophrenia may affect patient clinical responses; several factors can affect these levels, including patient medication adherence, body weight (BW) or body mass index, smoking habits, and sex. The cytochrome P450 (CYP) system is a major factor affecting the blood levels of antipsychotics because many antipsychotics are metabolized by this system. There were notable genetic differences between people of different races. In this study, we determined the racial or ethnic differences in the metabolic patterns of some selected antipsychotics by reviewing therapeutic drug monitoring studies in East Asian populations. The plasma concentrations of haloperidol, clozapine, quetiapine, aripiprazole, and lurasidone, which are metabolized by specific CYP enzymes, were determined to be higher, under the same daily dose, in East Asian populations than in Western populations.

## 1. Introduction

Psychotropic drugs, including antipsychotics, lithium, and antidepressants, have been prescribed for more than half a century. Empirical clinical studies have reported that East Asian patients may require lower dosages of psychotropic drugs, such as antipsychotics, lithium, and antidepressants, than non-Asians. For example, a study by Yamashita and Asano [1] indicated that Asian psychiatrists prescribed close to a half-dose of tricyclic antidepressants compared with an American study [2]. Another study by Takahashi et al. [3] demonstrated that Japanese patients appeared to require a lower effective therapeutic dose of lithium carbonate compared with their US counterparts. Furthermore, a controlled study by Okuma comparing the antimanic efficacy of carbamazepine and chlorpromazine observed that Japanese patients required a considerably lower mean therapeutic dose of chlorpromazine to reach an antimanic state than Western populations [4]. A study conducted by Lin and Finder [5] in the United States demonstrated that Asian patients with psychosis required significantly lower doses of neuroleptics than did Caucasian patients to control psychopathology to the same degree.

Pharmacokinetics is the study of the time course of drug release, absorption, distribution, metabolism, and excretion. Therapeutic drug monitoring (TDM) refers to the quantification and interpretation of drug plasma concentrations (Cps) to optimize pharmacotherapy. Both the pharmacokinetic and pharmacodynamic properties of a drug can affect the clinical response of a disease. The blood levels of antipsychotics used for the treatment of schizophrenia may affect their clinical responses; several individual factors can affect these levels, including patient medication adherence, body weight (BW) or body mass index, smoking habit, and sex. The cytochrome P450 (CYP) system plays a major role in moderating blood levels of psychotropic drugs. For example, CYP450S such as *CYP1A2, CYP3A4, CYP2C19*, and *CYP2D6* have the largest substrate populations and are responsible for the metabolism of the majority of psychotropic drugs and drug–drug interactions. Moreover, some antidepressants such as paroxetine and fluvoxamine are inhibitors, and the mood stabilizer carbamazepine induces the metabolism of some psychotropic drugs [6]. There are individual genetic differences in the CYP system and genotyping, such as in *CYP2D6* and *CYP2C19* enzymes, have become applicable to clinical practice [7].

Varying metabolic patterns in enzyme activity have been observed among different racial or ethnic populations. For example, Shimada et al. have compared the CYP enzymes in liver microsomes, including *CYP 1A2, 2A6, 2B6, 2C, 2D6, 2E1*, and *3A*, between 30 Japanese and 30 Caucasian patients and found that total CYP content was higher in Caucasian than in Japanese populations [8]. McGraw and Waller have reviewed the CYP450 variations in different ethnic populations, including *CYP1A2, 2C8, 2C9, 2C19, 2D6, 3A4,* and *3A5* single nucleotide polymorphisms (SNPs), and suggested that racial/ethnic differences in metabolic phenotypes could be explained by differences in SNP distribution [9].

This review evaluated racial/ethnic differences in the metabolic patterns of some antipsychotics, determined through TDM in East Asian populations.

## 2. Methods

We reviewed studies that performed TDM of the selected antipsychotics and included participants from East Asian countries such as China, Japan, Korea, Singapore, and Taiwan. Racial or ethnic differences in metabolic patterns were compared with those of Western populations. The reported Cps/daily dose (DD) was used as the main indicator of the comparison between races. As BW is a minor factor affecting the Cps of a medication, the mean weights of 70 and 85 kg for the Asian and Western populations, respectively, were used to obtain the standard ratio (85/70 = 1.21), and a difference of Cps ≥ 20% was considered significant.

## 3. Results

### 3.1. Haloperidol

The major and minor CYP enzymes for haloperidol metabolism are *CYP3A4* and *CYP2D6*, respectively [10]. Haloperidol is the most studied antipsychotic in the comparison of racial or ethnic differences. Potkin et al. [11] were the first to compare the Cps of haloperidol by using a standard BW-adjusted daily dose (0.4 mg/kg) between Chinese and American patients. They observed that the Chinese patients had a 52% higher mean Cps of haloperidol than did their matched American non-Asian counterparts when a daily dose of 0.4 mg per kilogram of BW was administered for 6 weeks. These results indicated that Asian patients require lower dosages of neuroleptic drugs than non-Asians. Moreover, the results demonstrated the higher sensitivity of Asians to neuroleptic-induced side effects.

Lin et al. [12] examined the Cps of haloperidol and prolactin in three groups of normal male volunteers (12 Caucasians, 11 American-born Asians, and 11 foreign-born Asians) after administering a single dose of haloperidol. The result revealed that the Caucasians had a lower Cps of haloperidol and a smaller prolactin response. The findings suggest that both pharmacokinetic and pharmacodynamic factors contribute to the differences in the response between Caucasians and Asians.

In previous findings of our haloperidol pharmacokinetic study [13], Taiwanese patients exhibited a significantly longer elimination half-life (25.5 ± 13.8 vs. 18.5 ± 6.6 h) and a larger area under the curve (AUC) concentration (221.1 ± 152.5 vs. 122.9 ± 31.8 ng × h/mL) after the administration of a single dose of 10 mg of haloperidol when compared with Western patients. Because the elimination half-life was more than 24 h, we then compared the Cps of haloperidol by using a cross-over design between a twice-a-day and once-a-day dosing regimen in the same patients and observed that a once-a-day dose of haloperidol before bed is adequate for most patients on maintenance therapy [14]. Our group also conducted a comparison study of the steady-state Cps of haloperidol and reduced haloperidol in Chinese (*n* = 156), Caucasian (*n* = 37), Hispanic (*n* = 51), and African-American (*n* = 23) populations [15]. We observed wide interpatient variability between haloperidol dose and Cps for each ethnic group, and the Chinese group had a higher Cps than the other ethnic populations.

### 3.2. Clozapine

Clozapine is extensively metabolized by CYP enzymes, including *CYP1A2* and *CYP3A4* [16], as well as *CYP2D6* and *CYP2C19* [17,18].

In a pharmacokinetic study, we administered a single dose of 100 mg of clozapine to 14 male patients with schizophrenia [19]. All patients were determined to be extensive metabolizers by a dextromethorphan probe. The mean elimination half-life was 13.7 ± 9.9 h, which is more than twice as long as those reported in previous studies from Germany (6.0 + 1.5 h) [20] and France (average of 7.6 h) [21].

In a study examining the steady-state Cps of clozapine by our group [22], the mean Cps/DD in 162 Taiwanese patients was 1.49 ng/mL/mg. We suggested that these patients had approximately 30 to 50% higher Cps than the Caucasian patients. A study conducted by Chong et al. in Singapore confirmed this finding [23].

Our group then analyzed sex differences in clozapine Cps and determined that women possessed a higher Cps (approximately one-third higher) of clozapine and norclozapine but not of N-oxide metabolites [24]. A Chinese study by Tang et al. also reported that 193 patients reported a higher Cps than that of Western populations, and the Cps was higher in female patients [25].

The mean Cps/DD in 131 Japanese patients reported by Yada et al. was 1.80 ± 0.79 ng/mL/mg, which was significantly higher than that in Caucasian patients [26].

These related studies were included in a critical review performed by de Leon et al. [27]. The authors suggested that Asian patients require only half the dose of clozapine usually prescribed for Caucasians. In addition, they indicated that laboratories that enable the routine use of clozapine should be developed. Moreover, the authors suggested that Asian pharmaceutical companies should consider developing clozapine formulations that allow the administration of lower doses, such as 12.5, 10, or 5 mg as starting doses for Asian patients.

### 3.3. Risperidone and Paliperidone

Risperidone is metabolized by *CYP 2D6* and partially by *CYP3A4* [28,29].

Risperidone is the most commonly used second-generation antipsychotic. Risperidone is metabolized into active metabolite 9-hydroxy risperidone (reduced risperidone), namely paliperidone. The ratio of risperidone to 9-hydroxy risperidone is a crucial indicator of the active moiety. As studies performing the TDM of risperidone measured both the parent compound and metabolite, we examined risperidone and paliperidone together.

#### 3.3.1. Risperidone

The relationship among CYP polymorphisms, plasma level, and therapeutic effects have been widely reported because *CYP 2D6* is a major enzyme for risperidone metabolism [30]. In this study, we reviewed studies comparing racial differences in risperidone metabolism. In the studies included in this review, blood sampling was performed 12 h after the evening dose. As not all studies reported the Cps in ng/mL, we estimated the data based on their results.

Heykants et al. [31] summarized the results of pharmacokinetic studies of risperidone in healthy adults or schizophrenic patients and reported that 6 mg/DD of risperidone resulted in 10 ng/mL of risperidone and 45 ng/mL of 9-hydroxyrispnidone, with the active moiety being 55 ng/mL. The Cps/DD was 9.1 ng/mL/mg, and the ratio was 0.22.

In Singapore, Lee et al. [32] performed a TDM analysis of risperidone in 20 patients with schizophrenia with a mean DD of 5.5 ± 1.6 mg. Their mean Cps of risperidone, 9-hydroxyrispnidone, and the active moiety was 9.1 ± 13.4, 38.7 ± 18.4, and 47.8 ± 29.7 ng/mL, respectively. The mean Cps/DD of the active moiety was approximately 8.6 ng/mL/mg, and the ratio was 0.24.

A study conducted by Spina et al. in Italy reported the relationship between the Cps of risperidone and 9-hydroxyrisperidone and examined the clinical response in 42 patients with schizophrenia [33]. The authors reported that the Cps of risperidone and its active metabolite correlated with the occurrence of parkinsonian side effects, whereas no significant correlation was noted with the degree of clinical improvement. The risperidone dosage at the completion of the study was 6.7 ± 1.3 mg/day, which resulted in a Cps of 19 ± 21 nmol/L for risperidone, 128 ± 68 nmol/L for 9-hydroxy-risperidone, and 146 ± 72 nmol/L for the active moiety. The mean Cps/DD of the active moiety was approximately 9.3 ng/mL/mg, and the ratio was 0.14.

A Taiwanese study by Lane et al. [34] examined the Cps of risperidone and 9-hydroxy risperidone in 31 patients treated with risperidone. They reported a Cps/DD of 9.5 ng/mL/mg for the active moiety, and a ratio of 0.19.

A Japanese study by Mihara et al. [35] examined 85 patients who were given a fixed dose of risperidone (6 mg/day), determining the Cps and genotyping the *CYP2D6* polymorphism. The results revealed that *CYP2D6* activity affects the Cps of risperidone by strongly regulating the 9-hydroxylation of risperidone. However, the presence of a similar active moiety among different genotype groups suggested that determination of the *CYP2D6* genotype has little importance in clinical situations. From their reports, we calculated that the mean Cps of the active moiety was approximately 6.6 ng/mL/mg, and the ratio was 0.11.

A Taiwanese study by Chen et al. [36] examined the correlation between scores on a continuous performance test and the Cps of risperidone in 10 patients with schizophrenia. The mean DD was 4.5 mg, and the Cps was 71.9 nmol/mL. We calculated the Cps of the active moiety per DD to be approximately 6.8 ng/mL/mg and the ratio to be 0.33.

A German study by Riedel et al. [37] examined 82 patients who were administered with risperidone with a mean dose of 4.3 ± 0.9 mg. The plasma levels of the active moiety (45.8 ± 27.4 ng) did not significantly differ between extensive metabolizers and poor metabolizers. Based on the mean daily dose, we calculated the Cps of the active moiety per DD to be 10.7 ng/mL/mg and the ratio to be 0.40.

A Japanese study by Yasui-Furukori et al. [38] evaluated the clinical response to risperidone by assessing the plasma Cps for a fixed dose of 6 mg of risperidone in 60 patients with schizophrenia. The mean active moiety Cps was 44.0 ± 27.8 ng/mL, which is equivalent to 7.3 ± 4.6 ng/mL/mg, and the ratio was 0.14.

Table 1 summarizes these results chronologically and the comparison between 5 studies from East Asian and 3 from Western populations. No significant differences in the active moiety Cps/DD and ratios were observed. Although a racial genetic difference was noted between *CYP2D6* and *CYP3A4*, the TDM of risperidone included the measurement of the parent compound and metabolite. Thus, this effect (risperidone plus 9-hydroxy risperidone) may not represent a significant difference between races.

#### 3.3.2. Paliperidone

Paliperidone is the active metabolite of risperidone, and the osmotic controlled release oral-delivery system can reduce maximum plasma peak level and fluctuation in Cps (Invega) [39]. Paliperidone is primarily excreted by the kidneys, and its clearance is decreased in patients with moderate to severe renal impairment.

In German, Nazirizadeh et al. [40] performed a TDM study of oral paliperidone in 106 patients. The mean Cps of paliperidone was 35.7 ± 25.2 ng/mL, with a mean DD of 7.8 ± 2.9 mg, and the mean Cps/DD was 4.7 ± 2.9 ng/mL/mg.

In Japanese patients, Suzuki et al. [41] evaluated 15 elderly patients with schizophrenia (63.4 ± 3.4 years), switching from risperidone to paliperidone. After 12 weeks of treatment, the mean dose of paliperidone was 6.2 ± 3.7 mg/day, and the mean Cps was 42.7 ± 17.2 ng/mL. When calculated by the stratification of different doses, the mean Cps/DD was 7.9/ng/mL/mg. As paliperidone is substantially excreted by the kidney, this higher number may have been due to the inclusion of older patients with compromised renal function.

A systematic review by Schoretsanitis et al. [42] reported a mean Cps/DD of 4.09 ng/mL/mg post-administration of oral paliperidone based on six studies, including a total sample of 221 non-Korean and non-geriatric adult patients, and they found a mean Cps/DD of 7.7 ng/mL/mg among 69 patients from three steady-state long-acting injectable (LAI) paliperidone monthly studies.

In Taiwan, our group compared the Cps of oral and LAI paliperidone in patients with schizophrenia [43]. The mean Cps per daily dose (Cps/DD) was 4.11 ± 1.99 ng/mL/mg in the oral group and 9.24 ± 3.78 ng/mL/mg in the LAI group. The oral results verified the report by Schoretsanitis et al. [42].

In Norway, Helland and Spigset [44] conducted a TDM study of LAI paliperidone in 185 patients, where patients were administered a median dose of 100 mg every 28 days. The median Cps/DD was 16.1 nmol/L/mg/, which was equivalent to 6.8 ng/mL/mg. This figure was approximately 25% lower than our result of 9.24 ng/mL/mg. However, this comparison was not adjusted for BW. When we back calculated the mean BW of our patients (66.9 kg) to 1.25-fold, the presumed mean BW of the study was 83.6 kg, which is acceptable in Western countries.

As paliperidone is a metabolite of risperidone, the first-pass metabolic effect of the CYP system affects paliperidone less than risperidone. In contrast to the findings of Western studies, no difference was observed in the BW-adjusted Cps/DD in TDM studies of oral or LAI paliperidone between East Asian and Western populations.

### 3.4. Olanzapine

Olanzapine is primarily metabolized by direct glucuronidation and *CYP1A2*, and, to a lesser extent, by *CYP2D6* and *CYP3A4* [45]. A pharmacokinetic study performed by Callaghan et al. [46] in healthy Caucasian, Japanese, and Chinese individuals revealed no racial differences in the metabolization of olanzapine. A study performed in Singapore by Sathirakul et al. [47] confirmed this finding by comparing healthy Chinese and Caucasian male individuals.

Lane et al. [48] conducted a pharmacokinetic study of a single dose (10 mg) of olanzapine in 21 male Taiwanese patients with chronic schizophrenia. The mean oral clearance and elimination half-life of olanzapine were 51.5 ± 61.6 L/h and 30.9 ± 4.3 h, respectively. The elimination half-life was comparable to that reported by Callaghan et al. (33 h) [46].

Bergemann et al. [49] conducted a TDM study in 71 German patients with schizophrenia. They observed that the mean daily dose of olanzapine was 17.5 ± 7.0 mg, and the mean olanzapine Cps was 54.2 ± 37.8 ng/mL. We estimated the Cps/DD to be 3.1 ng/mL/mg.

Lu et al. [50] analyzed the Cps of olanzapine in 151 Taiwanese patients with schizophrenia. The mean daily dose was 14.2 ± 5.4 mg, and the mean Cps of olanzapine was 37.0 ± 25.6 ng/mL, with Cps/DD equal to 2.9 ng/mL/mg. The authors indicated that that both female patients and nonsmokers had approximately a 50% higher Cps of olanzapine than their counterparts.

Overall, the results indicated no significant racial differences in the pharmacokinetics of olanzapine, as it is primarily metabolized by glucuronidation.

### 3.5. Ziprasidone

Ziprasidone is primarily metabolized by aldehyde oxidase and *CYP3A4* [51]. Vogel et al. [52] performed a retrospective analysis of ziprasidone in Germany. The TDM results revealed that the Cps did not differ between patients who received monotherapy and those who received multiple medications, indicating that ziprasidone may not be susceptible to pharmacokinetic drug–drug interactions. The mean daily dose was 144 mg, and the mean Cps was 78 ng/mL. We estimated the Cps/DD to be 0.54 ng/mL/mg. In a TDM study of ziprasidone in China [53], Lv et al. reported the Cps/DD to be 0.76 ng/mL/mg in Han Chinese patients and 0.64 ng/mL/mg in Mongolian patients. The findings of these studies indicate Chinese patients might have a higher Cps of ziprasidone (around 40%) than Westerns. Further studied are needed to confirm this finding.

### 3.6. Quetiapine

Quetiapine is mainly metabolized by *CYP3A4* [54] and partially by the *CYP**2D6* system [55]. Quetiapine is indicated for the treatment of schizophrenia and bipolar disorder, and as an adjunct treatment for major depressive disorder. It is also widely used in the treatment of other psychiatric disorders, such as major depression, generalized anxiety disorder, and insomnia.

A study conducted by Bakken et al. [56] in Norway reported that the mean Cps and Cps/DD of quetiapine was 0.56 nmol/L/mg (0.21 ng/mL/mg) in 927 samples obtained from 601 patients. In a study conducted by Fisher et al. [57] in the United Kingdom, the mean Cps of quetiapine was 195 ng/mL in 99 samples collected from 59 patients with a mean DD of 525 mg. We estimated the Cps/DD to be 0.37 ng/mL/mg. Another study performed by Handley et al. in the United Kingdom [58] examined TDM data from 946 samples obtained from 487 patients. The mean Cps was 215 ng/mL in 124 patients receiving 401–600 mg/day of quetiapine, and 296 ng/mL in 126 patients receiving 601–800 mg/day of quetiapine. We estimated the mean Cps/DD to be 0.42 ng/mL/mg in the two groups based on these dosages. A TDM study conducted by Hasselstrøm and Linnet [59] in Denmark on 62 patients (54 patients were co-medicated) revealed a median Cps/DD of 0.41 nmol/L/mg (0.16 ng/mL/mg). In the monotherapy of the quetiapine group (*n* = 8), the Cps/DD was 0.28 nmol/L/mg (0.11 ng/mL/mg).

Li et al. [60] conducted a TDM study in 21 Chinese patients receiving a fixed dose of 200 mg twice daily of quetiapine. The mean steady-state Cps of quetiapine was 147 ± 142 ng/mL, with the mean Cps/DD being 0.37 ng/mL/mg. Li et al. [61] monitored the Cps in patients receiving DD 300 (*n* = 13), 600 (*n* = 13), and 800 mg (*n* = 14) of quetiapine and found a mean Cps of 212, 320, and 551 ng/mL, respectively. We calculated the mean Cps/DD of the entire group to be 0.64 ng/mL/mg.

In Taiwan, our group performed a TDM study of 107 patients with various diagnoses [62]. The mean daily dose of quetiapine was 175.9 ± 184.4 mg (25–800 mg). The mean Cps of quetiapine was 105.6 ± 215.3 ng/mL, and the mean Cps/DD was 0.58 ± 0.55 ng/mL/mg.

These results indicate that Chinese individuals may have approximately 50 to 100 % higher Cps of quetiapine under the same DD.

### 3.7. Aripiprazole

Aripiprazole is mainly metabolized by the human CYP isozymes *CYP2D6* and *CYP3A4* [63].

In Taiwan, our group performed a TDM study of aripiprazole and its metabolite dehydroaripiprazole and found that the higher Cps of aripiprazole was correlated with clinical responses in patients with schizophrenia [64]. The Cps/DD in the study was 14.8 ± 6.0 ng/mL/mg for aripiprazole and 6.1 ± 2.4 ng/mL/mg for dehydroaripiprazole, which is similar to the results performed by Kim et al. [65] and Nakamura et al. [66], and was approximately 30 and 87% higher than those reported by Kirschbaum et al. [67] and Citrome et al. [68], respectively. These findings suggest an ethnic pharmacokinetic difference for aripiprazole and that approximately 50 to 100% of East-Asian patients have a higher Cps/DD than Western patients when body weight effects are considered.

### 3.8. Lurasidone

Lurasidone is mainly metabolized by *CYP3A4* [69].

Our group measured the Cps of lurasidone in 44 patients, with a mean dose of 82.7 ± 23.5 mg/day [70]. The mean plasma concentration 14–16 h after dosing was 15.3 ± 12.5 ng/mL, and the Cps/DD was 0.199 ± 0.188 ng/mL/mg.

In the United States, Findling et al. [71] conducted a pharmacokinetic study of lurasidone in children and adolescents and determined that it was similar to exposure observed at steady state in adults. They presented the mean steady state Cps of lurasidone in the group receiving 80 mg/day. The 16-h Cps was approximately 9.5 ng/mL by extrapolation, which was approximately 40% lower than those reported by our findings [70] (15.7 ± 12.8 ng/mL, *n* = 35).

In China, Zhang et al. [72] did a populational pharmacokinetic study of lurasidone and compared the racial differences between Chinese and Western populations. The typical predicted steady state AUC for lurasidone was 37% higher in Chinese participants (median: 264 ng × h/mL) compared with Western participants (median: 193 ng × h/mL).

These results indicate that Chinese individuals might have a 30 to 40% higher Cps/DD of lurasidone compared with their Western counterparts.

### 3.9. Brexpiprazole

Brexpiprazole is mainly metabolized by *CYP3A4* and *CYP2D6* enzymes [73].

A Japanese study by Ishigooka et al. [74] examining the pharmacokinetics of brexpiprazole revealed an elimination half-life of 52–92 h. Currently, no TDM study on brexpiprazole has demonstrated racial differences in the Cps/DD. As brexpiprazole is mainly metabolized by *CYP3A4* and *CYP2D6*, a racial difference could be possible, similar to other antipsychotics.

## 4. Discussion and Conclusions

Human genetic evolution leads to differences between individuals and between races, in terms of both genotype and phenotype. The CYP system is involved in drug metabolism and exhibits genetic differences in individuals, resulting in different levels of enzyme activity. McGraw and Waller [9] reviewed CYP variation in different ethnic populations looking at *CYP1A2, 2C8/9/19, 2D6*, and *3A4/5*, and suggested that genetic influences that drive racial/ethnic differences in phenotypic CYP450 metabolic activity have not been fully characterized. A better understanding of genetic determinants of the metabolic phenotype would help understand racial or ethnic differences and may eventually lead to the realization of personal medicine. Dorji et al. [75] reviewed *CYP2C9*, *CYP2C19, CYP2D6*, and *CYP3A5* polymorphisms in South-East and East Asian populations and concluded that an adequate number of studies have provided comparable results in these areas.

Clinically, TDM is performed to optimize medicine dosages. It is almost a clinical routine to monitor Cps when prescribing mood stabilizers such as lithium, valproic acid, or carbamazepine, but this is not the case with antipsychotics. de Leon et al. [76] suggested that the future of personalized medicine, with better pharmacokinetic and pharmacodynamic genetic testing, could ultimately lead to better clinical outcomes. Beyond individual genetic CYP functional differences, group racial/ethnic genetic CYP differences should not be overlooked. This review offers up-to-date information comparing the Cps of commonly used antipsychotics between East Asian and Western populations. Notably, the Cps of haloperidol, clozapine, ziprasidone, quetiapine, aripiprazole, and lurasidone, which are metabolized by specific CYP enzymes, were determined to be higher (30 to 100%) under the same daily dose in the East Asian populations. Table 2 presents the CYP enzyme system responsible for antipsychotic metabolism and the effects of racial differences.

In conclusion, in some antipsychotics metabolized by specific CYP enzymes, the Cps is higher in East Asian populations than in Western populations. Psychiatrists in the East Asian region need to be aware that smaller doses are required than suggested on the drug labels when prescribing these antipsychotics.

## Figures and Tables

**Table 1 jpm-12-01362-t001:** Summary of selected 5 publications from East Asian and 3 from Western countries in the therapeutic drug monitoring of risperidone.

Authors, Published Year (Country, *n*)	Risperidone/ 9-Hydroxyrisperidone Ratio	Active Moiety ng/mL/mg
Heykants et al. [31], 1994 (N. America, 178)	0.22	9.1
Lee et al. [32], 1999 (Singapore, 20)	0.24	8.6
Lane et al. [34], 2000 (Taiwan, 31)	0.19	9.5
Spina et al. [33], 2001 (Italy, 42)	0.14	9.3
Mihara et al. [35], 2003 (Japan, 85)	0.11	6.6
Chen et al. [36], 2004 (Taiwan, 10)	0.33	6.8
Riedel et al. [37], 2005 (Germany, 82)	0.40	10.7
Yasui-Furukori et al. [38], 2010 (Japan, 51)	0.14	7.3
Total (Mean ± SD)	0.22 ± 0.10	8.5 ±1.5
Ease Asian Mean (Mean ± SD)	0.23 ± 0.08	8.1 ± 1.2
Western Mean (Mean ± SD)	0.22 ± 0.13	8.9 ± 1.7

**Table 2 jpm-12-01362-t002:** Cytochrome P450 enzyme system responsible for antipsychotic metabolism and the effects of race.

Antipsychotics	Major Enzyme	Minor Enzyme	Minor Enzyme	Race effect of Cp: % E. Asian > Western
Haloperidol	CYP3A4	CYP2D6		50
Clozapine	CYP1A2	CYP3A4	CYP2D6, CYP2C19	30–50
Risperidone	CYP2D6	CYP3A4		--
Paliperidone	--	CYP2D6		--
Olanzapine	(Glucuronidation)	CYP1A2	CYP2D6, CYP3A4	--
Quetiapine	CYP3A4	CYP2D6		50–100
Ziprasidone	Aldehyde oxidase	CYP3A4		30-40
Aripiprazole	CYP2D6	CYP3A4		50–100
Lurasidone	CYP3A4			30–40
Brexpiprazole	CYP3A4	CYP2D6		NA

CYP, Cytochrome P450; Cp, Plasma Concentration; NA, not available.

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
