# Peer review of "Racial/Ethnic Differences in the Pharmacokinetics of Antipsychotics: Focusing on East Asians"

_jpm, 2022, doi:10.3390/jpm12091362_

Round 1

Reviewer 1 Report

I have reviewed the review article “Racial/Ethnic Differences in the Pharmacokinetic of Antipsychotics: Focusing on East Asians” by Shih-Ku Lin submitted for publication in the Journal of Personalized Medicine (MDPI).

My comments and questions to the author are below:

The manuscript is aimed to provide extensive knowledge of ethnic differences (Especially among East Asians) in the therapeutic doses of various antipsychotic drugs.

This review addresses one of the important contexts of the low-dose administration of drugs Haloperidol, Clozapine, Risperidone, Paliperidone, Olanzapine, Ziprasidone, Quetiapine, Aripiprazole, Lurasidone, and Brexpiprazole. In this review, the author brought to the attention with various clinical reports emphasizing the low dose administration of these drugs for East Asian patients when compared to the FDA-approved therapeutic dosage for Caucasians.

The overall review design and flow of the manuscript are straightforward, but it needs improvement in the following context.

1.       In the review author did not emphasize the genetic background behind the pharmacokinetics of these antipsychotics. It would be more appreciable if the author addresses the genetic variants among East Asians in these CYP enzymes and which play a major role in determining the dose requirement for these drugs.  This will provide a strong interest to the readers of the review and will lead to developing any future pharmacogenetics screening panel specific for the East Asian population to be screened prior to initiating the drug treatment.

2.       References 13, 14, and 64 are the self-citations of the author’s previous articles. It would be better if the author mentions them as “in our previous findings (Ref)” instead of Lin et al (Ref). This will make the readers know the author is a pioneer researcher in the field for decades.

3.       The discussion and conclusion need to be more elaborate with more ideas and knowledge outcomes of the current review. This part should be well written for providing or filling the knowledge gap in the current research field.

Author Response

  1. In the review, author did not emphasize the genetic background behind the pharmacokinetics of these antipsychotics. It would be more appreciable if the author addresses the genetic variants among East Asians in these CYP enzymes and which play a major role in determining the dose requirement for these drugs. This will provide a strong interest to the readers of the review and will lead to developing any future pharmacogenetics screening panel specific for the East Asian population to be screened prior to initiating the drug treatment.

A: I will add more information and references in this aspect.

The cytochrome P450 (CYP) system is a major factor affecting the blood levels of psychotropic drugs. For example, CYP450S such as CYP1A2, CYP3A4, CYP2C19, and CYP2D6 have the largest substrate population and are responsible for the majority of psychotropic drugs metabolism, and drug-drug interactions. Moreover, some antidepressants such as paroxetine and fluvoxamine are inhibitors, and mood stabilizer carbamazepine is an inducer to the metabolism of some specific psychotropic drugs [6]. Individual genetic differences occur in CYP system and genotyping such as in CYP2D6 and CYP2C19 enzymes have become applicable in clinical practice [7].

Varying metabolic patterns in enzyme activity have been observed among different racial or ethnic populations. For example, Shimada et al. have compared the CYP enzymes in the liver microsomes including CYP 1A2, 2A6, 2B6, 2C, 2D6, 2E1, and 3A between 30 Japanese and 30 Caucasians and found the total CYP content was higher in Caucasian than in Japanese populations [8]. McGraw and Waller have reviewed the CYP450 variations in different ethnic populations including CYP1A2, 2C8, 2C9, 2C19, 2D6, 3A4, and 3A5 single nucleotide polymorphisms (SNPs), and suggested that racial/ethnic differences in metabolic phenotype can be explained by differences in SNP distribution [9].

  1. References 13, 14, and 64 are the self-citations of the author’s previous articles. It would be better if the author mentions them as “in our previous findings (Ref)” instead of Lin et al (Ref). This will make the readers know the author is a pioneer researcher in the field for decades.

A: Thank you for your suggestion and I will revise the content accordingly. Please refer to the revised version words in red.

  1. The discussion and conclusion need to be more elaborate with more ideas and knowledge outcomes of the current review. This part should be well written for providing or filling the knowledge gap in the current research field.

A: I will revise the content accordingly. Please refer to the revised version words in red.

        Clinically, TDM is performed to optimize a medicine dosage. It is almost a clinical routine to monitor Cps when prescribing mood stabilizers such as lithium, valproic acid, or carbamazepine, but this is not the case with antipsychotics. de Leon et al. [76] suggested that the future of personalized medicine, with better pharmacokinetic and pharmacodynamic genetic testing, could ultimately lead to better clinical outcomes. Beyond these individual genetic CYP functional differences, the group racial/ethnic genetic CYP differences should not be overlooked. This review offers the current information comparing the Cps of clinically popular used antipsychotics between Ease Asian and Western populations. It is noticeable that the Cps of haloperidol, clozapine, quetiapine, aripiprazole, and lurasidone, which are metabolized by specific CYP enzymes, were determined to be higher (30% to 100%) under the same daily dose in the East Asian population. Table 2 presents the CYP enzyme system responsible for antipsychotic metabolism and the effects of racial differences.

In conclusion, in some antipsychotics metabolized by specific CYP enzymes, the Cps is higher in East Asian populations than in Western populations. Psychiatrists in the East Asian region need to be aware that smaller doses are required than suggested on the drug labels when prescribing these antipsychotics.

Reviewer 2 Report

In this review, the data of the therapeutic drug monitoring studies concerning antipsychotics used for the treatment of patients with schizophrenia and bipolar disorder in  East Asian countries (China, Japan, Korea, Singapore, Taiwan)  and in Western countries were compared to confirm racial/ethnic differences between the two populations.  The plasma level of a relevant drug after its daily dose was evaluated. Obtained results indicate that with an equivalent drug dose the East Asian patients exhibit for some antipsychotics  30-100 % higher plasma levels which means they require lower dosages of these neuroleptic drugs than do non-Asians. Moreover, the higher drug plasma level in East Asians was associated with a higher incidence of side effects.

The general conclusion is that racial/ethnic genetic differences in cytochrome P450 responsible for the metabolism of xenobiotics are implicated.

Abstract and the Discussion & Conclusion should be revised and rewritten.  The abstract should contain the background, material & method, results, and conclusion. Noteworthy, out of  10 examined drugs five were metabolized slowly.

Author Response

Thank you for your kind review. The abstract will be kept unchanged as other articles’ formats in this special issue.

https://www.mdpi.com/2075-4426/12/7/1155/htm